# Harnessing and Distilling ChatGPT's Ability to Bridge Semantic Variance for Precise Query-Document Alignment in Encephalitis Research: Surpassing Keyword-Based Search Engines

**Santosh Gupta**
SanGupta.ML@gmail.com

## ABSTRACT

Keyword-based search engines often fail in retrieving information that aligns with user query intent, due to variations in keywords and phrasing in scientific literature. This paper introduces an encephalitis query-document dataset, characterized by its high semantic variability. Our dataset comprises thousands of query-document pairs. To represent the diverse linguistic expressions found in encephalitis research, we leveraged the advanced language understanding capabilities of GPT-4 to generate queries that, while conceptually aligned with the information in the documents, significantly differ in phrasing and terminology. This approach addresses a critical need in scientific literature searches – retrieving pertinent information that might be overlooked due to conventional keyword-based search limitations.

To evaluate the efficacy of our dataset, we trained a specialized transformer model capable of converting these query-document pairs into embeddings. Our results demonstrate a significant improvement in retrieving relevant encephalitis research papers, especially those that are not surfaced by traditional search engines like PubMed. This enhanced retrieval performance not only underscores the potential of embedding-based retrieval in medical literature search, but also opens up new avenues for comprehensive literature exploration. The implications of our findings extend beyond encephalitis studies, suggesting broader applicability for similar methodologies in other specialized fields of research.

*Keywords*— Transformers, Information Retrieval, Embeddings, GPT, Encephalitis.

## 1. INTRODUCTION

Encephalitis research, like many medical fields, suffers from information retrieval challenges, often due to the semantic gap in traditional search engines. The field of encephalitis research, akin to many areas within the medical sciences, is fraught with information retrieval (IR) challenges that significantly impede the progress of research and clinical understanding. Encephalitis, characterized by the inflammation of the brain, presents a wide spectrum of causes, manifestations, and treatments, making the search for accurate and relevant scientific literature a daunting task. Traditional search engines, often fall short when tasked with bridging the semantic gap inherent in scientific queries and the documents that contain their answers. This discrepancy arises from the diverse linguistic expressions authors may use to describe similar concepts, coupled with the highly specific and nuanced nature of medical terminology. Consequently, researchers and clinicians face substantial hurdles in accessing literature that precisely matches their inquiry, potentially overlooking critical insights that reside beyond the reach of keyword-based search algorithms.

In recent years, lots of research has been going towards embeddings based retrieval to address the limitations of keyword based searches [1]. In this paper, we describe an approach of leveraging the language understanding capabilities of GPT-4 to create a semantically varied encephalitis query-document dataset.

## 2. DATASET AND TRAINING METHODOLOGY:

Our dataset, accessible on Hugging Face [2], includes 53.1k rows of PMC passages and PubMed abstracts, Each with several queries. The release dates of the papers spans January 2000 to July 2023. Utilizing the ChatGPT API, we created semantically varied queries for each text passage and abstract in our dataset, ensuring a wide representation of linguistic expressions.

The dataset was trained over using a contrastive loss function, with a batch size of 32, over 10 epochs. Each document in the dataset is associated with several queries, and one was selected at random to create each query-document pair. The base model used is the ncbi/MedCPT-Article-Encoder transformer from the National Center for Biotechnology Information [3]. The resultant retrieval model, hosted on Hugging Face [4], demonstrates improvements over traditional search methods.

A Google Colab demo [5] , which downloads the models and papers then loads and sets up the search system, can be used by anyone to perform queries on 513k PMC passages and 47.7k PubMed abstracts, whose release dates range from January 2000 to December 2023.

## 3. PERFORMANCE EVALUATION AND CASE STUDIES

Our model's performance, evaluated against traditional PubMed searches, demonstrates its ability to retrieve relevant information that PubMed fails to retrieve. This includes instances where our model identified significant research papers and insights from queries where PubMed yielded zero results. Detailed examples of these

comparative analyses are provided in the Appendix, illustrating the model's capacity in handling semantically varied queries.

## 4. CONCLUSION

Our model demonstrates a significant improvement over traditional keyword-based searches. By leveraging the semantic variance understanding of large language models for dataset creation, we offer an model for encoding and retrieving encephalitis research. Our approach not only improves access to encephalitis literature but also suggests a broader applicability for similar methodologies in other fields facing challenges in semantic variance and information retrieval.

## LINKS

[2] Encephalitis Query-Document Dataset: https://huggingface.co/datasets/Santosh-Gupta/EncephalitisQueryDocuments

[4] Encephalitis Retrieval Model: https://huggingface.co/Santosh-Gupta/EncephalitisRetrieval

[5] Encephalitis Retrieval Demo: https://colab.research.google.com/drive/1wN1a32DWCKmP3mgPw7GEJq9I54PSMh7b?usp=sharing

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

## APPENDIX: CASE STUDIES

These case studies provide an in-depth look at the performance of our model compared to PubMed's default search, highlighting its ability to find relevant information that keyword-based methods might overlook. Each case analysis the PMC and the Pubmed abstract results separately, since they were indexed separately by the encoder.

### Case Study 1: Chronic Loss of Endothelial Nitric Oxide and APP Pathology

**Query**: "Chronic loss of endothelial nitric oxide (NO) as a contributor to amyloid precursor protein (APP) related pathology."

**Default Pubmed Search Results**: 0 results. (an altered version of this query returned 1 result 'Endothelial nitric oxide deficiency promotes Alzheimer's disease pathology').

**Model Results**:

PMC Embeddings, 1st Result:

-Paper: "Use of Eflornithine (DFMO) in the Treatment of Early Alzheimer's Disease: A Compassionate Use, Single-Case Study." https://pubmed.ncbi.nlm.nih.gov/29559907/ [6]

-Paragraph Retrieved: "Most animal models of AD involve transgenic expression of mutated amyloid precursor protein (APP), which leads to parenchymal Aβ deposition but does not usually invoke tau pathology and associated neuronal cell death. Consequently, these models have been criticized as incomplete models of AD pathology (Irizarry et al., 1997; Radde et al., 2008). In a recently developed CVN-AD mouse model, immune-mediated nitric oxide (iNOS) was lowered to mimic human levels, resulting in a model that demonstrates the complete pathological course of AD, including parenchymal amyloidosis, gradual spread of hyperphosphorlyated tau, episodic memory impairment, and significant hippocampal neuronal degeneration (Colton et al., 2008, 2014; Wilcock et al., 2014). These mice are mNos2-deficient and transgenic for the Swedish K670N/M671L vasculotropic Dutch/Iowa E693Q/D694N mutant (APPSwDI) APP (Kan et al., 2015), which prevents the expression of inducible nitric oxide synthase protein, thus lowering the level of nitric oxide production (for a detailed description of this model, see Colton et al., 2008, 2014; Wilcock et al., 2014)."

-Relevancy Analysis: Highly relevant to the query. It explains that in a CVN-AD mouse model, immune-mediated nitric oxide (iNOS) was lowered to mimic human levels, which resulted in a model that accurately replicates the complete pathological course

of Alzheimer's disease (AD), including amyloidosis and tau pathology, and this reduction in nitric oxide production is achieved by preventing the expression of inducible nitric oxide synthase protein. This connection highlights the relevance of endothelial nitric oxide (NO) loss as a contributor to amyloid precursor protein (APP) related pathology in the context of AD research."

PMC Embeddings, 2nd Result:

-Paper: "The blood–brain and gut–vascular barriers: from the perspective of claudins." https://pubmed.ncbi.nlm.nih.gov/34152937/ [7]

-Paragraph Retrieved: "Alzheimer's disease is a chronic neurodegenerative disease that causes severe dementia that is characterized by memory loss, impaired reasoning, and personality alterations. The pathological hallmarks of this disease include the formation of amyloid beta (Aβ) plaques in the brain parenchyma and around blood vessels and neurofibrillary tangles in the neurons, which are composed of hyperphosphorylated tau proteins.109,110 Histological analysis of postmortem human AD brains has shown BBB breakdown as defined by plasma albumin and immunoglobulins around amyloid plaques.111 In 2019, Nation and colleagues presented evidence of BBB impairment by measuring soluble platelet-derived growth factor receptor-β (sPDGFRβ) as a biomarker of damaged capillaries in patients with AD. However, they showed the circulation of sPDGFRβ by cerebrospinal fluid independent of amyloid plaques and tau status by positron emission tomography, which suggested vascular dysfunction as a component of AD pathology.112 Aβ toxicity and neuroinflammation have been associated with down-regulation of tight junction proteins in patients with AD who suffer from cerebral amyloid angiopathy at the capillary level. Two studies by Carrano and colleagues showed that in postmortem brains of patients with cerebral amyloid angiopathy, at the capillary level there was a remarkable reduction, or even complete loss, of claudin-5, among the other tight junction proteins. The capillaries were positive for Aβ plaques or were associated with activated microglia that were positive for nicotinamide adenine dinucleotide phosphate oxidase 2, an enzyme that is responsible for reactive oxygen species production.113,114 Although Viggars et al. correlated increased levels of albumin and fibrinogen to more progressed AD pathology in human postmortem brains, the expression levels of claudin-5, occludin and ZO-1 were not affected.115 Nevertheless, BBB impairment is well accepted as a pathological characteristic of human AD pathology.112"

-Relevancy analysis: Highly relevant to the query.Describes various aspects of Alzheimer's disease (AD) pathology, including the formation of amyloid beta (Aβ) plaques, which are directly related to amyloid precursor protein (APP) pathology. Although it does not directly mention endothelial nitric oxide (NO), it discusses blood-brain barrier (BBB) impairment and vascular dysfunction in AD, which could be implicitly related to NO levels since NO is known to play a significant role in vascular health and function. The passage suggests that vascular issues are a component of AD pathology, indirectly linking to the potential role of chronic loss of endothelial NO in APP-related pathology in AD.

Abstract Embeddings, 1st Result:

-Paper: "Endothelial nitric oxide deficiency promotes Alzheimer's disease pathology." https://pubmed.ncbi.nlm.nih.gov/23745722/ [8]

-Paragraph Retrieved: "Aging and the presence of cerebrovascular disease are associated with increased incidence of Alzheimer's disease. A common feature of aging and cerebrovascular disease is decreased endothelial nitric oxide (NO). We studied the effect of a loss of endothelium derived NO on amyloid precursor protein (APP) related phenotype in late middle aged (LMA) (14-15\u00a0month) endothelial nitric oxide synthase deficient (eNOS(-/-) ) mice. APP, \u03b2-site APP cleaving enzyme (BACE) 1, and amyloid beta (A\u03b2) levels were significantly higher in the brains of LMA eNOS(-/-) mice as compared with LMA wild-type controls. APP and A\u03b21-40 were increased in hippocampal tissue of eNOS(-/-) mice as compared with wild-type mice. LMA eNOS(-/-) mice displayed an increased inflammatory phenotype as compared with LMA wild-type mice. Importantly, LMA eNOS(-/-) mice performed worse in a radial arm maze test of spatial learning and memory as compared with LMA wild-type mice. These data suggest that chronic loss of endothelial NO may be an important contributor to both A\u03b2 related pathology and cognitive decline. Cardiovascular risk factors are associated with increased incidence of Alzheimer's disease (AD). A common feature of these risk factors is decreased endothelial nitric oxide (NO). We observed, in mice deficient in endothelial nitric oxide synthase, increased amyloid precursor protein (APP), \u03b2-site APP cleaving enzyme 1, amyloid beta levels, microglial activation, and impaired spatial memory. This suggests chronic loss of endothelial NO may be an important contributor to the pathogenesis of sporadic AD."

-Relevancy Analysis: Highly relevant to the query. Discusses the impact of a chronic loss of endothelial nitric oxide (NO) on amyloid precursor protein (APP) related phenotype in mice. Paragraph cover. Same as only Pubmed result for "Endothelial nitric oxide deficiency promotes Alzheimer's disease pathology "

Abstract Embeddings, 2nd Result:

-Paper: "Pharmacological strategies for the regulation of inducible nitric oxide synthase: neurodegenerative versus neuroprotective mechanisms." https://pubmed.ncbi.nlm.nih.gov/16765486/ [9]

-Paragraph Retrieved:  "Inducible nitric oxide synthase (iNOS) is one of three NOS isoforms generating nitric oxide (NO) by the conversion of l-arginine to l-citrulline. iNOS has been found to be a major contributor to initiation/exacerbation of the central nervous system (CNS) inflammatory/degenerative conditions through the production of excessive NO which generates reactive nitrogen species (RNSs). Activation of iNOS and NO generation has come to be accepted as a marker and therapeutic target in neuroinflammatory conditions such as those observed in ischemia, multiple sclerosis (MS), spinal cord injury (SCI), Alzheimer's disease (AD), and inherited peroxisomal (e.g. X-linked adrenoleukodystrophy; X-ALD) and lysosomal disorders (e.g. Krabbe's disease). However, with the emergence of reports on the neuroprotective facets of NO, the prior dogma about NO being solely detrimental has had to be modified. While RNSs such as peroxynitrite (ONOO(-)) have been linked to lipid peroxidation, neuronal/oligodendrocyte loss, and demyelination in neurodegenerative diseases, limited NO generation by GSNO has been found to promote vasodilation and attenuate vascular injury under the same ischemic conditions. NO generated from GSNO acts as second messenger molecular which through S-nitrosylation has been shown to control important cellular processes by regulation of expression/activity of certain proteins such as NF-kappaB. It is now believed that the environment and the context in which NO is produced largely determines the actions (good or bad) of this molecule. These multi-faceted aspects of NO make therapeutic interference with iNOS activity even more complicated since complete ablation of iNOS activity has been found to be rather more detrimental than protective in most neurodegenerative conditions. Investigators in search of iNOS modulating pharmacological agents have realized the need of a delicate balance so as to allow the production of physiologically relevant amounts of NO (such as those required for host defence/

neutotransmission/vasodilation, etc.) but at the same time block the generation of RNSs through repressing excessive NO levels (such as those causing neuronal/tissue damage and demyelination, etc.). The past years have seen a noteworthy increase in novel agents that might prove useful in achieving the aim of harnessing the good and blocking the undesirable actions of NO. It is the aim of this review to provide basic insights into the NOS family of enzymes with special emphasis of the role of iNOS in the CNS, in the first part. In the second part of the review, we will strive to provide an exhaustive compilation of the prevalent strategies being tested for the therapeutic modulation of iNOS and NO production."

-Relevancy Analysis: Highly relevant to the query. Discusses the role of inducible nitric oxide synthase (iNOS) in generating NO and its implications in various central nervous system (CNS) conditions, including neurodegenerative diseases like Alzheimer's disease (AD).

## Case Study 2: Intracranial Antigens and Glymphatic System

**Query**: "Intracranial antigens interaction with immune system though glymphatic system."

**Default Pubmed Search Results**: 0 results.

**Model Results:**

PMC Embeddings, 2nd Result:

-Paper: "Meningeal lymphatic drainage promotes T cell responses against Toxoplasma gondii but is dispensable for parasite control in the brain." https://pubmed.ncbi.nlm.nih.gov/36541708/ [10]

-Paragraph Retrieved: "In other organs, lymphatic vessels serve as conduits for the transport of tissue-derived antigen and dendritic cells to lymph nodes, where naive and memory T cells are optimally positioned for detection of their cognate antigen (Thomas et al., 2016; Gasteiger et al., 2016). The recent discovery of functional lymphatic vessels in the dura mater layer of meninges has prompted a significant reconsideration of how the CNS engages the peripheral immune system (Louveau et al., 2015b; Aspelund et al., 2015). Meningeal lymphatic vessels are observed in rodents, primates, and humans (Absinta et al., 2017; Albayram et al., 2022), and in experimental models of brain cancer and autoimmunity these vessels have been shown to play an integral role in regulating T cell responses in the CNS (Song et al., 2020; Louveau et al., 2018b). Mouse studies have demonstrated that meningeal lymphatic vessels convey macromolecules and immune cells from the meninges and cerebrospinal fluid (CSF) to the deep cervical lymph nodes (Louveau et al., 2018b). Indeed, when model antigens like ovalbumin (OVA) are injected into the brain, these molecules travel from the brain interstitium into the CSF via glymphatic flow (Iliff et al., 2012) and have the potential to be presented to T cells in the deep cervical lymph nodes (Ling et al., 2003; Harris et al., 2014)."

-Relevancy Analysis: Very relevant, quote " Indeed, when model antigens like ovalbumin (OVA) are injected into the brain, these molecules travel from the brain interstitium into the CSF via glymphatic flow (Iliff et al., 2012) and have the potential to be presented to T cells in the deep cervical lymph nodes (Ling et al., 2003; Harris et al., 2014)."

## Case Study 3: Visual Field-Cut and Immune-Mediated Encephalitis

**Query**: "Visual field-cut as a potential symptom of immune-mediated encephalitis."

**Default Pubmed Search Results**: 1 result.

**Model Results**:

PMC Embeddings, 1st Result:

-Paper: "Autoimmunity in visual loss." https://pubmed.ncbi.nlm.nih.gov/27112687/ [11]

-Paragraph Retrieved: "There are also less numerous, but nonetheless significant, projections to other visual cortical areas (such as V5 involved in motion processing) and the tectum (involved in pupillary reflexes). There is, therefore, an ample substrate for differential effects of immune processes on parallel pathways of processing within a single anatomically defined structure such as the retina, optic nerve, or visual cortex. Selective deficits of color vision, motion perception, and other modalities are all potential manifestations of autoimmune-mediated dysfunction.Fig. 20.1The major cell types of a typical mammalian retina. From the top row to the bottom, photoreceptors, horizontal cells, bipolar cells, amacrine cells, and ganglion cells. For steric reasons, only a subset of the wide-field amacrine cells is shown. (Reprinted from Masland (2001) with permission from Macmillan Publishers (Nature Neuroscience)).Table 20.1Definition of terms used in visual sciencesTermDefinitionHard-wiredIn human vision the first-, second-, and third-order neurons and their axons are hard-wired into the human brain and transmit analogue and digital signals. This hard-wired single pathway enables the retinotopic map of the human visual cortex. There is no (or very little) potential for plasticity in the strict definition of this single pathway model (Balk et al., 2015)Analogue signalThe analogue signal produced by the photoreceptors is continuous. The signal intensity varies over time depending on the light-induced metabolism of opsins. Therefore the variation of the signal carries information on light entering the eye (Fig. 20.2). The analogue signal of the photoreceptors is converted by retinal bipolar cells into a digital signal as required for higher-level visual network processingDigital signalThe digital signal of the hard-wired visual pathway is sampled from the analogue signal fed into retinal bipolar cells by photoreceptors. The digital signal consists of a series of action potentials. The information of the digital signal is encoded in the time frequency of these action potentialsRetinotopicA topographic map where adjacent locations on the retina are represented by adjacent neurons in the dorsal lateral geniculate nucleus and V1.Rayleigh criterionThe minimum resolvable detail according to the generally accepted physical definition, diffraction limitation. Simplified, the limitation of image resolution relates to the order of wavelength of the wave used to image it. For example, the Rayleigh criterion for a wavelength of 500 nm and a circular pupil opening of 5 mm is: $\theta R = 1.22 \times \lambda d = 1.22 \times 5 \times 10{-}5 cm 0.5 cm = 1.22 \times 10{-}4 rad$. Put into relation, a Snellen acuity of 6/6 (UK notation, 20/20 US notation) corresponds to a resolution limitation of $\theta = 5 \times 10$–4rad in most subjects. Under optimal circumstances a visual acuity of $\theta = 2 \times 10$–4rad might be achieved. Essentially, visual acuity depends on the anatomic spacing of sensory neurons in the retina and the wavelengths of the light entering the eyeVernier acuityThe human visual cortex can make spatial distinctions with a precision which is about 10 times better than visual acuity. This so-called hyperacuity depends on sophisticated information processing in the visual human brain. Vernier acuity represents the quintessential example of hyperacuity where the alignment of two edges or lines can be judged with a better precision than predicted by visual acuity. Clinically, the assessment of, for example, normal stereopsis relies on hyperacuity"

-Relevancy Analysis: Highly relevant to the query. Discusses how immune processes can differentially affect visual processing pathways in structures like the retina, optic nerve, or visual cortex.

PMC Embeddings, 4th Result:

-Paper: "Biologic Therapy in Refractory Non-Multiple Sclerosis Optic Neuritis Isolated or Associated to Immune-Mediated Inflammatory Diseases. A Multicenter Study." https://pubmed.ncbi.nlm.nih.gov/32796717/ [12]

-Paragraph Retrieved: "Optic neuritis (ON) is an acute inflammatory optic neuropathy that may be associated with dramatic visual loss and an important decrease in quality of life in absence of an adequate treatment. Multiple Sclerosis (MS) ON, the most common form of presentation, is characterized by unilateral acute retroocular pain and visual loss, more commonly observed in Caucasian women between 18 and 50 years [1]. Visual acuity (VA) in patients with MS-ON usually improves within a few months even without treatment [2,3,4]. Non-MS ON is less frequent and can be an isolated disorder or related to infections and immune-mediated diseases such as Neuromyelitis Optica (NMO) or other systemic diseases [5]. Non-MS ON may have atypical features such as male gender, age less than 18 or greater than 50 years, absence of pain and bilateral presentation [5]. In non-MS ON, a chronic progressive disease is more common. Flare-ups are frequent, leading often to visual loss [3,6]. If not promptly treated, the visual outcome can be devastating, causing a severe visual loss, and even with adequate treatment, many patients may worsen over months [7,8,9,10]."

-Relevancy Analysis: Highly relevant to the query. Discusses immune-mediated encephalitis in the context of optic neuritis (ON), a related inflammatory condition affecting the optic nerve.