# OpenReview forum: "Harnessing and Distilling ChatGPT's Ability to Bridge Semantic Variance for Precise Query-Document Alignment in Encephalitis Research: Surpassing Keyword-Based Search Engines"
_AAAI.org/2024/Spring_Symposium_Series/Clinical_FMs — AAAI 2024 SSS on Clinical FMs_

### Official Review · Reviewer_Qsco · 2024-02-15
**Ineffectiveness of naive baselines on tricky datasets is widely known**

**Rating:** 5
**Confidence:** 4

**Review:**

Perhaps the more useful contribution of this paper is in leveraging GPT-4 to create synthetic data in niche domains that can be tailored for specific purposes (in this case, benchmarking models that have poor keyword overlap between the document and query).
However, the points regarding shortcoming of keyword based models is already very well known, but also in this case quite expected since the dataset is explicitly constructed to fool keyword matching. It would make more sense to not cast the retrieval model's performance as proof for a move towards embedding-similarity based retrieval (as that has been the case for many years now) but as proof that the dataset is interesting. If that is the intention, it is more beneficial to provide principles and details of the prompt used in the ChatGPT API that enabled the curation of this dataset, so that a practitioner can then apply such principles to their own use case. That would be a simple two page report detailing the challenges and innovations required and the iteration cycles that one may have to go through, along with simple strategies for checking the quality of the generated dataset.

---

### Official Review · Reviewer_yZdn · 2024-02-19

**Rating:** 6
**Confidence:** 4

**Review:**

Summary: The paper introduces an approach to enhance literature retrieval from PubMed for encephalitis-related research. The approach is based on a transformer-based embedding model that, given encephalitis-related queries, retrieves relevant PubMed literature. Unlike prior keyword-based searching systems, the proposed method is able to identify articles relevant to a query even when the exact keywords are absent from the query. To build the embedding model, the authors first created a dataset of thousands of query-document pairs. Specifically, they used ChatGPT to generate queries from the document as it has the ability to understand the variance of encephalitis representations and capture various semantics of the document.

Comments and suggestions:

1. Comparing the performance to that of a keyword-based searching system seems insufficient. Would it be possible to use a pre-trained, high-quality embedding model to perform this task? I believe recent advancements in retrieval-augmented generation (RAG) would provide useful techniques for this task.
2. How do we ensure that a general-purpose model like ChatGPT understands encephalitis literature, as I suppose it belongs to long-tail distribution?
3. The link for [1] is missing.
4. It is stated that the "Biopython" library is used. However, the "Biopython" library I know of is a biological sequencing tool (https://biopython.org/), which is not related to this paper.
5. The paper would benefit from a large-scale quantitative assessment of the model's performance. This addition would provide a clearer understanding of its efficacy compared to existing methods.
6. I think adding a graphical diagram would be beneficial to better outline the pipeline.

---

### Official Review · Reviewer_wybD · 2024-02-21

**Rating:** 6
**Confidence:** 4

**Review:**

This paper introduces an encephalitis query-document dataset along with an embedding model used for retrieval. The experimental results demonstrate the superiority of the embedding based model over traditional key-based search engines.

**Pros:**

(1) The dataset and the model are valuable to the community.

(2) The code is clear and easy to read.

**Cons:**

(1) The experimental results only present some cases, instead of the performance on the whole dataset. It would be better to list a series of numbers in a table. For example, for the baselines, choose keyword-based search engines, as well as OpenAI’s text-embedding APIs. The metrics can be recall, etc. Also, the authors can explore whether the cheaper model developed by users (like the one proposed in this paper) can be on par with the OpenAI text-embedding API models.

(2) The layout of this paper can be further improved. For instance, move the “Links” parts as the footnotes. For the case study paragraphs in the appendix, it would be better to wrap them with a blockquote. It would be more convenient to use latex templates to modify these things, compared with Word.

(3) The title is too long and can be shorten, such as “Enhancing Encephalitis Research Retrieval: Leveraging GPT-4 for Semantic Query-Document Alignment Beyond Keywords.”

---

### Official Review · Reviewer_WE7Z · 2024-02-24

**Rating:** 6
**Confidence:** 4

**Review:**

This paper addresses the issue that general-purpose search engines have inaccurate retrieval results for medical field content, especially for encephalitis research. Thus, they introduce a query-document set based on PubMed data. They further use GPT-4 to generate queries query varies that conceptually aligned but with different phrases and terms. A model is trained on the introduced data with contrastive loss. Results show the retrieval capabilities are enhanced.

Pros:
* The motivation is strong and work on the task is urgently needed
* Dataset is sampled, the model is trained, and results show the model works. The entire lifecycle is demonstrated with sufficient information

Cons:
* Now info on the collected dataset is limited, it would be great to show the data distribution and provide some samples
* Evaluation result is brief, more interpretation and analysis are appreciated
* How the contrastive loss is calculated, and how the data variants are used for the loss calculation needs to be clarified